# The Assessment of Green Business Environments Using the Environmental–Economic Index: The Case of China

**Cheng-Wen Lee [1,]*[ID], Chin-Chuan Wang [2], Hui-Hsin Hsu [2] and Peiyi Kong [3][ID]**

1   Department of International Business, Chung Yuan Christian University, Taoyuan 320, Taiwan
2   Ph.D. Program in Business, College of Business, Chung Yuan Christian University, Taoyuan 320, Taiwan
3   School of Economics and Management, Xiamen Nanyang University, Xiamen 361012, China
*   Correspondence: chengwen@cycu.edu.tw

**Abstract:** The quality of a country's business environment speaks volumes about its government's capacity and competitiveness. Unfortunately, the current system only evaluates countries and cities, overlooking the business environments of individual provinces. To address this issue, this study utilizes a green and sustainable development approach to evaluate the business environments of 30 provinces/municipalities in China. By incorporating ecological and environmental protection and sustainable development indicators, a novel green business environment index is constructed and analyzed to determine its impact on macroeconomic sustainable development and micro-enterprise operation. Taking into account the business environment index established by the World Bank and other organizations, this evaluation system adds ecological and environmental indicators specific to each province/municipality in China from the year 2011 to 2020. The result is a provincial green business environment evaluation index system consisting of 5 primary indicators and 30 secondary indicators. Principal component analysis (PCA) is then applied to rank the green business environment for each province/municipality. Furthermore, the overall green business environment of the Eastern region is superior to that of the Central and Western regions, highlighting the uneven development of the business environment in China.

**Keywords:** green business environment; sustainable economic development; Spatial Dubin Model

## 1. Introduction

The UN's 2030 Agenda emphasizes the importance of promoting high-quality economic growth through the establishment of a green business environment (GBE). Achieving a balance between economic and environmental concerns requires a deep understanding of the sustainable development process. The concept of GBE is broad, and its theoretical consensus is still being developed. It is often associated with Green GDP, Green and Low Carbon Economy, Green and Sustainable Development, and Green Total Factor Productivity. Research shows that a greener business environment enhances a country's growth potential. For instance, Zhao et al. used China's solid waste rate as a measure of its ecological baseline and found that a greener business environment boosts economic growth [1].

Optimizing and protecting the corporate environment serve different, but equally vital, purposes. The business environment evaluation index was considerably improved after the National Development and Reform Commission (NDRC) incorporated the ecological environment as an evaluation indicator in 2020. Generally speaking, one of the subsystems reflecting the business environment is the ecological environment [2]. The greatest strategy to maintain the business climate is to build the economy on the foundation of protecting the natural environment. In line with this perspective, Song and Mei suggest that ecological welfare performance measures the relationship between ecological resource inputs and welfare outputs, reflecting the sustainable development status of a region [3].

Creating a healthy balance between the market and the government is crucial for a successful economy. Alongside laws, regulations, supervisory systems, and financial

support policies, it is important to implement green support policies like researching and developing green technology, digitizing the green industry, and enforcing laws and regulations. The government plays a critical role in directing, administering, and ensuring the success of the green economy, while businesses develop green technology and make informed decisions with the help of the government's business platform. This leads to less pollution and better energy conservation. To achieve a GBE, the entire society must work together, including the government, businesses, and enterprises. Industries and businesses should establish GBE strategies considering the synergy of economic gains and environmental protection [4].

This study introduces the GBE Evaluation Index and incorporates the notion of sustainable and green development into the existing business environment evaluation index for future studies. The GBE emphasizes rational resource management and environmental protection to achieve sustainable development and high-quality economic growth, rather than simply combining green development and the business environment. Optimizing the creation of a GBE can benefit market participants' commercial behavior, increase enterprise competitiveness, promote enterprise transformation, and coordinate the high-quality, low-carbon development of enterprises. In China, a GBE assessment index with Chinese characteristics should be developed to measure the business environment along with the current business environment evaluation index.

## 2. Literature Review

### 2.1. Sustainable Development Goals (SDGs) Compliance

The United Nations identified 17 Sustainable Development Goals (SDGs) in 2015, which encompass the three dimensions of sustainable development: social, economic, and environmental. This study considers economic development, foreign investment, fixed asset investment, the ability to raise capital, financing capacity, and transport efficiency as indicators of the economic environment and population, inflation, disposable income, employment, social security level, and wage level as indicators of the social environment, as related works [5–7] have suggested. Achieving the SDGs is a global responsibility, and a healthy business climate can accelerate a resilient, inclusive, and sustainable recovery. The study also highlights the significance of renewable energy, land use, digital technologies, and high-quality education. Quality education indicators include inputs of education, higher education, and cultural atmosphere, while health insurance and social security level indicators are also vital. Clean energy indicators focus on power consumption, land use indicators consider land cost, and digital technology indicators focus on enterprise digitization.

### 2.2. Green Economic Perspective

The concepts of green economy, green growth, and green development have gained significant attention globally. The United Nations defines a green economy as promoting sustained, inclusive, and sustainable economic growth; full and productive employment; and decent work for all. Fair distribution is a word that comes up frequently in the context of the green economy [8]. Thus, we use "wage level" as the metric with which to measure fair distribution.

The green growth approach, according to the OECD, aims to promote economic growth and prosperity while preserving the availability of natural resources and environmental services that are vital to our well-being. Investment, innovation, and jobs should be the catalysts for new kinds of economic growth [9]. Therefore, inventions and patents, technological input, and technological innovation are grouped as technology and innovation indicators, while employment is used as an employment indicator.

In October 2008, the "Global Green New Deal" initiative was introduced as a response to the economic crisis of the time. This initiative gave rise to the idea of green development theory. The theory suggests that nations should prioritize long-term development when creating economic stimulus plans, establishing green institutions, promoting sustainable

development, and working towards global green improvements [10]. China formally introduced the concept of green development in 2015, with the release of the white paper "China's Green Development in the New Era" in 2023. This publication reflected the emergence of ecological civilization. Green development frequently involves industry and towns [11], and this study examines the effects of GBE measures on green development, with GDP being a key macroeconomic indicator of industrialization.

### 2.3. Environmental, Social, and Governance (ESG) Framework

Currently, more and more businesses are not only considering macroeconomic and environmental factors, but also focusing on eco-friendly operations. A prime example of this is the emergence of environmental, social, and governance (ESG) evaluation indicators. Chinese scoring agencies like Wind, MSCI, FTSE Russell, Sustainalytics, etc., have developed ESG evaluation indicators. ESG assesses environmental factors, such as wastewater, exhaust emissions, and energy usage, to determine how a company is performing in terms of sustainability [12]. Additionally, Song and Mei surveyed 30 provinces, autonomous regions, and municipalities using the environmental resource consumption and human development index [3]. Thus, we adopt green environmental indicators, including power consumption, environmental protection expenditure, waste disposal, air pollution, and living environment, in this research.

With the growing importance of the ecological economy and green environmental conservation, many organizations have reevaluated their business philosophies and models. To ensure long-term growth and profitability, companies now recognize the need to adopt green mindsets and business models. This involves considering internal and external factors, prioritizing shareholders and employees, taking on social responsibility, and operating sustainably. ESG is a framework used to evaluate a company's performance in these areas.

This research introduces the GBE assessment index, which is a combination of previous evaluation indices and a new ecological environment evaluation index that is tailored to Chinese provinces. The existing evaluation indices do not account for essential factors that affect the business environment, such as education and technological progress. The GBE Evaluation Index includes five components: economic environment, government environment, social environment, technical environment, and green environment.

### 2.4. Related Theories of Government Governance

The business environment plays a crucial role in the sustainable economic and social development of a country, and is an essential indicator of its overall strength and competitiveness. It also reflects the government's ability to govern efficiently. Evaluating the business environment helps to determine not only how well it is performing in a particular economy, but also how effectively the government is fulfilling its public service obligations [12]. For China, establishing an appropriate evaluation index system is vital for accurately assessing the efficiency of local government governance [13]. Indicators like government revenue scale, government balance, and tax are used to assess the government environment.

### 2.5. Green Business Environment Evaluation Index (GBEEI)

The development balance between Chinese provinces and cities is uneven, as per the country's overall recovery. Long-term globalization is now leaning towards regionalization or localization [14,15]. Bigger cities with more external access experience a slower recovery, while their surrounding areas bounce back more quickly. Improving the business environment is now a top priority, and the government's resilience is critical during this time. To achieve this, the government must concentrate on increasing social and psychological expectations, boosting development confidence, enhancing consumption, and improving the business environment. This aligns with the current vision of economic development in China, which was explored in the GBE study.

In China, the government is prioritizing the improvement of the business environment to support sustainable growth. However, the current literature only offers evaluation indicators from either a business environment or sustainable development perspective, without combining the two. Furthermore, the available indicators for assessing the business environment are found mostly at the national level or in international cities, resulting in a lack of provincial indicators with which to compare and identify gaps. This also makes it challenging for the government to enhance the business environment for better corporate satisfaction.

The data utilized in this study are public and accessible. The "wage level" indicator utilizes data from the Economic and Social Development Statistics Bulletin (2011–2020) of each province, whereas the "capability to acquire capital" and "financing capacity" indicators utilize data from the Wind Financial Terminal Database. The rest of the data were sourced from the China Statistic Yearbook (2012–2021). Table 1 shows the description of the green business environment (GBE) evaluation index, including 29 indicators.

**Table 1.** The green business environment evaluation index.

| Dimensions | No. | Indicators | Indicator Description | Indicator Explanation |
|---|---|---|---|---|
| Economic Environment | X11 | Economic Development | Gross regional product index (Previous year = 100) | Econometric development level |
| | X12 | International Trade | Total amount of import and export of goods (USD 1000) | Level of foreign trade |
| | X13 | Foreign Investment | Actual use of foreign direct investment (USD 10,000) | Increased confidence in the investment environment |
| | X14 | Fixed Asset and Investment | Fixed asset investment price index (previous year = 100) | Whether the enterprise is optimistic about future economic development |
| | X15 | Enterprise Digitization | The proportion of enterprises with e-commerce transaction activities (%) | The degree of digitalization of the enterprise |
| | X16 | Capability to Acquire Capital | Number of listed companies | The quantity and quality of listed companies determine the economic scale and height of a province |
| | X17 | Financing Capacity | Social financing scale | Economic attractiveness and financing capacity of a province |
| | X18 | Transport Efficiency | Cargo turnover (billion ton-kilometers) | Logistics development status |
| Government Environment | X21 | Government Revenue Scale | The ratio of local general budget revenue to GDP (CNY ten billion) | The government's ability to improve the quality of the provinces' business environments via financial support |
| | X22 | Government Balance | Local government's general budget revenue minus public service expenditure (%) | The ability of the government to coordinate the stable development of the economy |
| | X23 | Tax | The ratio of tax revenue to GDP (%) | The economic status of each province |
| | X24 | Land Cost | The ratio of land purchase cost to land purchase area (CNY/square meter) | The land cost of the enterprise |

| Dimensions | No. | Indicators | Indicator Description | Indicator Explanation |
|---|---|---|---|---|
| Social Environment | X31 | Population | Urban population density (person/square kilometer) | Provinces' economic attractiveness |
| | X32 | Inflation | Consumer price index (previous year = 100) | Purchasing power |
| | X33 | Disposable Income | Per capita disposable income of residents (CNY) | The wealth of the people |
| | X34 | Employment | Urban registered unemployment rate (million people) | Measures slack labor capacity |
| | X35 | Social Security Level | Number of participants in basic medical insurance (million people) | People's living standards |
| | X36 | Wage Level | Monthly minimum wage standard of each province (The highest grade, CNY) | Reflects social employment and income thresholds |
| Technical Environment | X41 | Input of Education | Education expenditure divided by local general public budget expenditure (CNY 100 million) | Related to the quality of citizens and the long-term development of the country |
| | X42 | Higher Education | The number of colleges and universities or institutions | Conducive to personnel training |
| | X43 | Inventions and Patents | Number of effective invention patents (pieces) | Conducive to the progress and development of science and technology |
| | X44 | Technology Input | Technology market turnover per unit of GDP (%) | Transformation and upgrading of economic structure |
| | X45 | Technological Innovation | The number of new product development projects (pieces) | Promotion of social development |
| | X46 | Cultural Atmosphere | Public library holdings per capita ((books per person)) | Improvement of humanistic quality |
| Green Environment | X51 | Power Consumption | Electricity consumption per CNY 100 million of GDP | Consumption of resources and the environment |
| | X52 | Environmental Protection Expenditure | Environmental protection expenditure in local government fiscal expenditure (CNY 100 Million) | Degree of protection of resources and environment |
| | X53 | Waste Disposal | Harmless treatment capacity of municipal solid waste (10,000 tons) | Reduces environmental pollution and waste of resources |
| | X54 | Air Pollution | Emissions of $SO_2$, $NO_2$, CO, O, $PM_{10}$, and $PM_{2.5}$ | Degree of pollution to the environment Chinese National Ambient Air Quality Standards (CNAAQS) |
| | X55 | Living Environment | Parks and green areas per capita (square meters/per person) | Improvement of human living environment |

## 3. Methodology

### 3.1. Principal Component Analysis

In the areas of economics, management, geography science, and other fields, principal component analysis (PCA) has been widely used to analyze non-random and random variable data [9]. Additionally, the PCA approach has been used in the field of the business

environment because of its qualities and the advantage of producing a comprehensive score based on many indications [16]. The PCA method involves using statistical techniques to analyze a set of criteria and extract common components [17]. By identifying representative elements in multiple criteria, PCA achieves dimension reduction. Additionally, it groups similar criteria into major components to test the relationships between variables. PCA reduces the complexity of interconnected variables or indices by creating a new combination with fewer, and unrelated, indicators or variables. To accomplish this, the PCA process involves several steps. The first step is variable standardization, which employs the z-score approach, expressed as follows in Equation (1).

$$Z_{ij} = \frac{X_{ij} - \overline{X}_j}{\sigma_j} \tag{1}$$

In order to assess the suitability of variables for PCA, the Kaiser–Meyer–Olkin (KMO) index and Bartlett's test of sphericity are commonly utilized. This study specifically employed Bartlett's test of sphericity with a significance level of $p < 0.001$ to examine the inter-correlation among variables, while the KMO index was used to measure sample adequacy and evaluate the validity of the PCA. The subsequent steps involved calculating the number of primary components based on their variance contribution rates and eigenvalues, determining the expression for the linear combination of primary components, and using the eigenvalues as weights to compute the composite index. To simplify the evaluation of the green business environment in provinces and municipalities, this thesis adopted the PCA method to reduce the number of indicators to 29 and obtain an overall score.

### 3.2. Spatial Econometrics

The first rule of spatial geography states that the strength of the correspondence between features is affected by distance. The components are more closely linked when the geographic distance is less [18]. Additionally, it is evident that the business climate and regional development are spatially correlated. However, traditional economic models are biased and do not consider spatial effects. Therefore, due to the variation in the business climate across provinces and municipalities in China, which may impact spatial effects, a spatial econometric model was used based on previous research [19]. In this study, two components, namely, the spatial correlation test and spatial econometric modeling, are utilized for the spatial econometrics procedure. Before implementing the spatial econometric model, it is necessary to measure the spatial dependence of the explained and explanatory variables. The investigation in this thesis uses both global space autocorrelation and local space autocorrelation as perspectives. The Global Moran's I is presented in this manner (Equation (2)).

$$\text{Moran's I} = \frac{n\sum_{i=1}^{n}\sum_{j=1}^{n} w_{ij}(x_i - \overline{x})(x_j - \overline{x})}{S^2 \cdot \left(\sum_{i=1}^{n}\sum_{j=1}^{n} w_{ij}\right)} \tag{2}$$

In this study, a Moran index scatterplot is utilized to display the local spatial autocorrelation. The empirical investigation of this paper utilizes the economic distance weight matrix, as it can indicate the spatial correlation between provinces with high economic proximity. Additionally, the geographic distance weight matrix is used for the robustness test to prevent any bias in the findings from a single matrix estimate. Based on the above analysis, the economic distance weight matrix ($W_1$) and geographic distance weight matrix ($W_2$) are constructed in the following manner (Equations (3) and (4)):

$$W_1 = \begin{cases} \dfrac{1}{\frac{1}{n}\left|\sum_{2011}^{2020} PGDP_i - \sum_{2011}^{2020} PGDP_j\right|2}, & i \neq j \\ 0, & i = j \end{cases} \tag{3}$$

$$W_2 = \begin{cases} \dfrac{1}{d_{ij}}, & i \neq j \\ 0, & i = j \end{cases} \tag{4}$$

For modeling purposes, the regional per capita GDP (*PGDP*) and the number of years (*n*) are represented by *PGDP* and *n*, respectively. The distance between the geographic centers of regions *i* and *j* is represented by $d_{ij}$. It is important to normalize both spatial weight matrices before the modeling process. To describe spatial impacts, researchers often use the Spatial Lag Model (SLM) and the Spatial Error Model (SEM) after examining spatial correlation. The Spatial Dubin Model (SDM) is a typical form of both the SLM and the SEM [18]. The SDM analyzes the effects of neighboring independent variables on their own dependent variables, as well as the spatial spillover effect of dependent variables in adjacent regions. Although the SDM is well-known for analyzing spatial spillover effects, its effectiveness requires further investigation. The following are general representations of the SDM (Equation (5)):

$$Y_{it} = \rho W Y_{it} + \alpha X_{it} + \theta W X_{it} + \xi_{it} + \varphi_{it} + \varepsilon_{it} \tag{5}$$

This study aims to use spatial econometric models to determine if whether there is a spatial effect of the GBE in the 30 provinces and municipalities of China. In the models, $\alpha$ represents the corresponding parameter and denotes the random error term. Here, $\rho$ and $\theta$ are spatial regressive coefficients; *W* represents the spatial weight matrix; and $\xi_{it}$ and $\varphi_{it}$ represent spatial and time effects, respectively. If $\theta = 0$, then SDM is equivalent to SLM. If $\theta + \rho\alpha = 0$, then SDM is equivalent to SEM.

## 4. Empirical Result

### 4.1. Result of Principal Component Analysis

To determine whether the variables are suitable for the principal component analysis (PCA) method, a computation process was performed using the KMO Index and Bartlett's Test. The KMO statistical value for the years 2011–2020 was greater than 0.60, while the significance of Bartlett's test for all years was less than 0.05. These results indicated that the data were appropriate for PCA. To identify the main components, we used the data-driven principle and applied the PCA method. By combining all the indicators, we obtained a better understanding of the overall situation. After examining the variance contribution and conducting the applicability test for PCA, we found that five components had initial eigenvalues greater than 1, which meant that they could explain 84.54% of the total variables. Therefore, we chose these five components for further analysis.

To calculate the GBE Evaluation Index in China from 2011 to 2020, we used the principal components' eigenvalues as weights to determine the composite index. Table 2 displays the results of this process. By averaging the scores of each province over the ten-year period, we found that GD, SH, JS, BJ, and ZJ were the top 5 provinces and municipalities out of 30. On the other hand, XJ, GZ, NX, GS, and QH ranked as the lowest five provinces and municipalities.

**Table 2.** The GBE scores for provinces and municipalities (2011–2020).

| ID No. | Province (PR) | | 2011 | 2012 | 2013 | 2014 | 2015 | 2016 | 2017 | 2018 | 2019 | 2020 |
|---|---|---|---|---|---|---|---|---|---|---|---|---|
| 1 | Beijing | BJ | 69.61 | 75.96 | 73.39 | 69.86 | 76.65 | 72.53 | 81.16 | 80.04 | 70.73 | 72.58 |
| 2 | Tianjin | TJ | 53.20 | 50.95 | 49.25 | 54.83 | 57.13 | 50.81 | 48.64 | 45.62 | 49.55 | 45.61 |
| 3 | Hebei | HEB | 42.70 | 44.38 | 43.29 | 41.04 | 41.19 | 41.52 | 42.97 | 44.28 | 46.28 | 46.99 |
| 4 | Shanxi | SX | 37.31 | 37.39 | 36.39 | 33.43 | 34.63 | 33.36 | 34.99 | 35.99 | 37.63 | 37.10 |
| 5 | Neimenggu | NM | 41.49 | 40.74 | 38.61 | 33.58 | 37.25 | 33.45 | 35.76 | 36.90 | 35.41 | 36.26 |
| 6 | Liaoning | LN | 56.30 | 57.89 | 50.68 | 49.16 | 44.11 | 39.79 | 38.58 | 43.95 | 41.34 | 39.10 |
| 7 | Jilin | JL | 36.26 | 36.24 | 33.82 | 33.94 | 35.72 | 32.06 | 33.15 | 35.38 | 32.83 | 32.58 |
| 8 | Heilongjiang | HL | 35.59 | 39.36 | 34.77 | 31.51 | 32.63 | 32.23 | 30.56 | 32.85 | 36.21 | 29.58 |
| 9 | Shanghai | SH | 77.55 | 76.35 | 72.91 | 82.16 | 81.91 | 85.18 | 80.91 | 79.13 | 84.57 | 82.12 |
| 10 | Jiangsu | JS | 75.16 | 75.18 | 78.64 | 79.70 | 73.13 | 72.38 | 72.12 | 74.92 | 69.99 | 76.73 |
| 11 | Zhejiang | ZJ | 67.33 | 60.35 | 71.02 | 72.06 | 66.97 | 68.50 | 68.95 | 71.18 | 68.79 | 73.72 |
| 12 | Anhui | AH | 43.67 | 41.57 | 41.71 | 42.71 | 42.08 | 44.18 | 44.80 | 42.33 | 41.02 | 42.03 |
| 13 | Fujian | FJ | 43.52 | 42.08 | 43.79 | 43.60 | 42.71 | 45.18 | 44.39 | 40.51 | 45.15 | 44.03 |
| 14 | Jiangxi | JX | 34.24 | 35.35 | 36.90 | 36.97 | 38.78 | 40.14 | 37.70 | 38.07 | 39.59 | 39.48 |
| 15 | Shandong | SD | 57.98 | 56.12 | 61.09 | 56.91 | 56.15 | 56.61 | 55.87 | 59.40 | 54.22 | 59.40 |
| 16 | Henan | HA | 41.52 | 42.77 | 41.78 | 40.98 | 42.14 | 43.30 | 41.50 | 43.71 | 44.97 | 44.68 |
| 17 | Hubei | HUB | 42.55 | 41.58 | 40.55 | 42.58 | 42.57 | 46.52 | 44.02 | 41.69 | 41.82 | 40.31 |
| 18 | Hunan | HUN | 39.46 | 38.20 | 39.08 | 40.26 | 39.48 | 41.96 | 38.64 | 38.71 | 39.92 | 40.02 |
| 19 | Guangdong | GD | 79.93 | 80.73 | 91.45 | 89.70 | 85.57 | 89.60 | 88.75 | 95.60 | 91.48 | 98.58 |
| 20 | Guangxi | GX | 33.59 | 33.62 | 33.10 | 35.33 | 32.87 | 34.10 | 33.27 | 33.15 | 33.43 | 31.82 |
| 21 | Hainan | HN | 34.79 | 34.73 | 31.00 | 34.91 | 34.35 | 40.87 | 37.35 | 33.33 | 35.40 | 34.67 |
| 22 | Chongqing | CQ | 38.82 | 39.42 | 37.98 | 37.31 | 36.64 | 37.66 | 38.77 | 36.29 | 36.40 | 36.64 |
| 23 | Sichuan | SC | 41.20 | 44.03 | 45.50 | 42.58 | 42.46 | 44.28 | 42.86 | 42.37 | 44.04 | 43.66 |
| 24 | Guizhou | GZ | 32.06 | 32.84 | 31.07 | 32.29 | 33.61 | 33.01 | 35.84 | 33.08 | 34.72 | 33.59 |
| 25 | Yunnan | YN | 34.27 | 35.36 | 35.23 | 36.06 | 36.36 | 35.68 | 34.33 | 32.64 | 34.85 | 31.36 |
| 26 | Shaanxi | SN | 35.42 | 37.49 | 37.04 | 36.35 | 36.80 | 36.55 | 38.52 | 36.55 | 39.60 | 37.59 |
| 27 | Gansu | GS | 27.95 | 28.67 | 28.25 | 27.97 | 32.11 | 28.93 | 31.42 | 30.27 | 30.79 | 30.92 |
| 28 | Qinghai | QH | 29.16 | 26.79 | 27.86 | 28.99 | 32.33 | 29.11 | 28.80 | 27.15 | 26.57 | 25.84 |
| 29 | Ningxia | NX | 33.13 | 27.50 | 29.92 | 29.67 | 29.85 | 28.52 | 33.36 | 32.10 | 29.02 | 30.84 |
| 30 | Xinjiang | XJ | 34.23 | 36.35 | 33.93 | 33.55 | 31.80 | 31.99 | 32.01 | 32.79 | 33.64 | 32.16 |

*4.2. Spatial Correlation Test Results*

From 2011 to 2020, Moran's I value for sustainable economic development was consistently positive and passed a significant level test of 5%. This suggests that there are significant spatial agglomeration and spillover effects associated with the spatial distribution of regional sustainable economic development and the Green Business Environment. To verify the spatial clustering characteristics of the Global Moran's I, the Local Moran's I was used. Since there was a large amount of data, a Moran scatter plot was used to show the Local Moran's I. In the Local Moran's I scatter plot (Figure 1), the first quadrant represents a High–High agglomeration area (HH), the second quadrant is a High–Low agglomeration area (HL), the third quadrant indicates a Low–Low agglomeration area (LL), and the fourth quadrant is a Low–High agglomeration area (LH). A High–High agglomeration area means that the sample itself is high and the sample around it is also high, while a Low–Low agglomeration area means the sample itself is low, with a low sample neighborhood around it.

There are examples of the Local Moran scatter plot that display the spatial clustering of regional sustainable economic development in the years 2011 (Figure 2a) and 2020 (Figure 2b). It is clear that the majority of provinces (municipalities) have their regional sustainable economic development concentrated in the first quadrant (High–High agglomeration areas) and the third quadrant (Low–Low agglomeration areas), indicating a significant, positive spatial correlation. The numbers shown in Figure 2a,b correspond to each province's ID number, as referenced in Table 2.

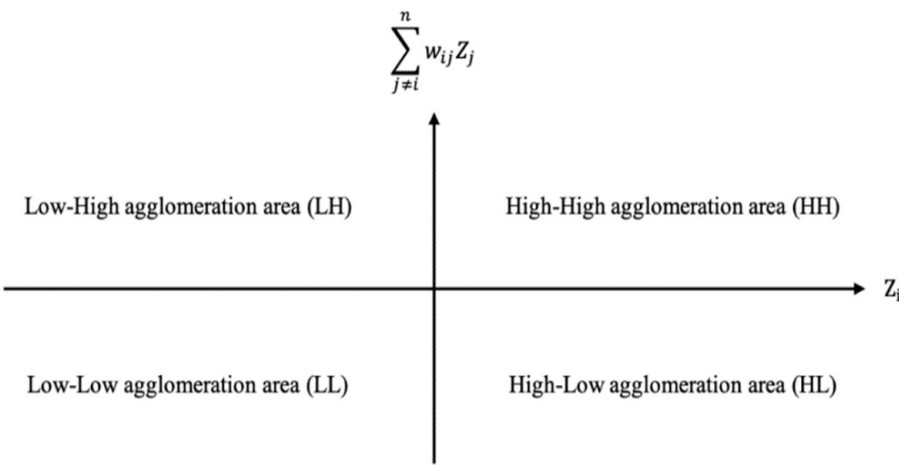

**Figure 1.** Schematic of the local Moran's I's four quadrants.

According to the results, we can find examples of the Local Moran scatter plot that display the GBE Evaluation Index spatial clustering for the years 2011 and 2020, represented in Figure 3a,b, respectively. Let us focus on Figure 3a. The first quadrant illustrates provinces with high GBE scores surrounded by other provinces with similarly high GBE scores, while the third quadrant shows lower GBE scores surrounding provinces with equally low GBE scores. By analyzing the province ID number, it becomes evident that the Eastern region in China is primarily concentrated in the "High–High" agglomeration areas, whereas the Central and Western regions are predominantly located in "Low–Low" agglomeration areas.

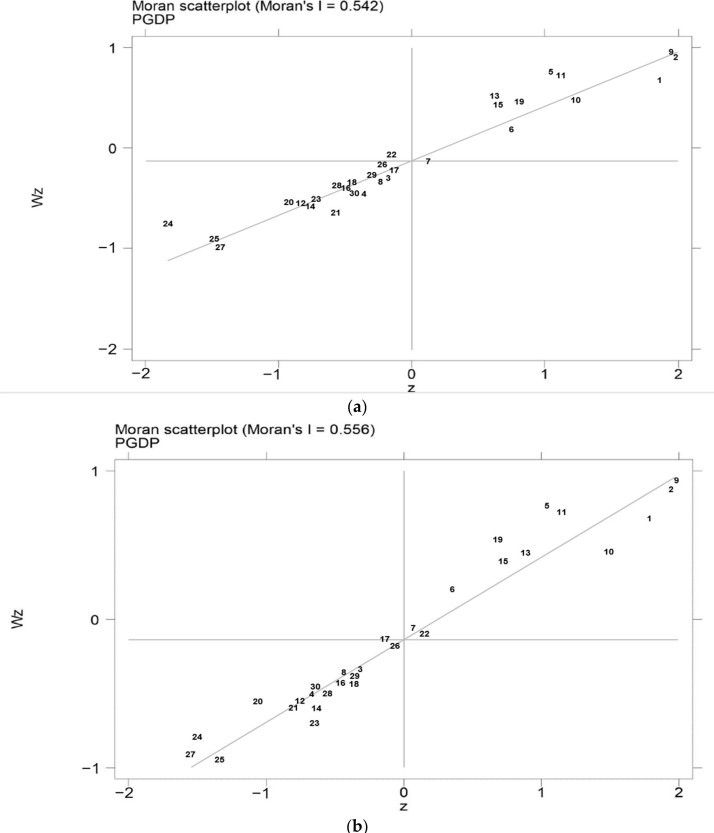

**Figure 2.** (**a**) Local Moran scatter plot for regional sustainable economic development (2011). (**b**) Local Moran scatter plot for regional sustainable economic development (2020).

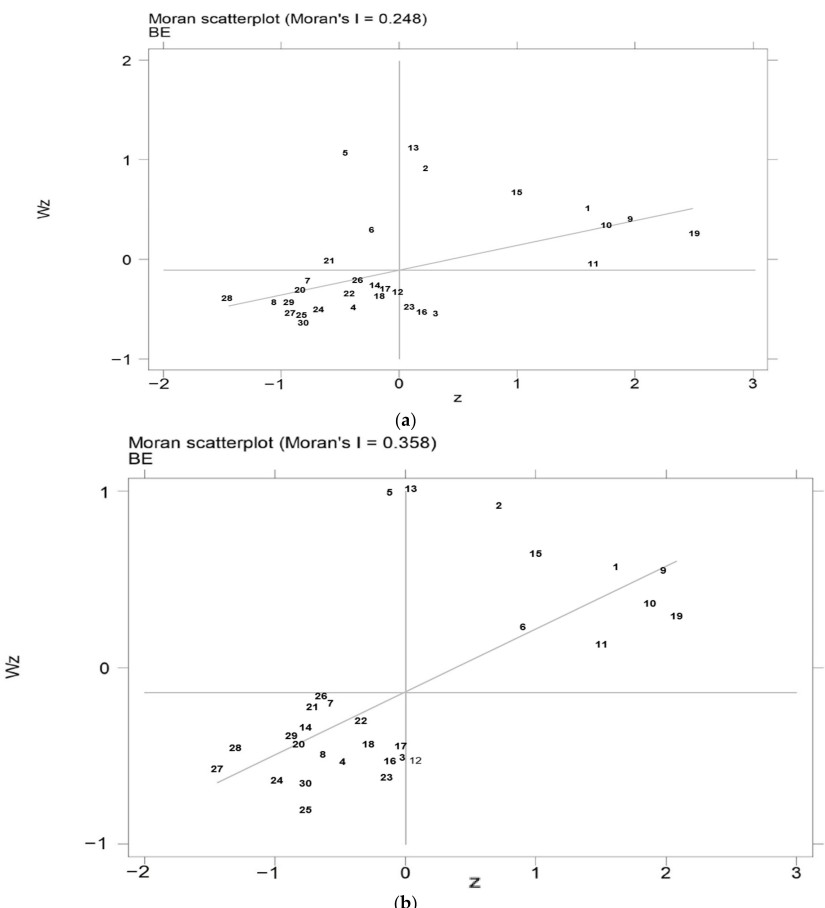

**Figure 3.** (**a**) Local Moran scatter plot for regional GBE evaluation index (2011). (**b**) Local Moran scatter plot for regional GBE evaluation index (2020).

*4.3. Statistical Test Results for Model Selection*

This study examines the impact of the green business environment on sustainable economic development in different areas. The spatial panel model has been divided into two distinct effect models, namely, fixed-effect and random-effect, based on differences in residual component decomposition. To determine the best model for the analysis, we refer to the results of Hausman's test. The null hypothesis was rejected at a significance level of 1%, which is indicated by the statistic in Table 3. This means that the fixed-effect model is the optimal choice. If the statistical software renders a *p*-value of 0.000, it means that the value is very low, with many instances of "0" before any other digit.

Secondly, based on the results of the log-likelihood (LR) test, it is evident that the double-fixed model is appropriate, as the null hypothesis regarding the joint significance of spatial fixed effects is rejected. Additionally, both the robust LM spatial lag and the robust LM spatial error models pass the significance test, indicating that both the spatial lag and spatial error models can be applied effectively.

Thirdly, in a spatial regression analysis, SDM, SLM, and SEM can be utilized. However, it is important to identify the model that best fits the data. The applicability of SDM was tested using the Wald and LR tests. The estimated results in Table 4 demonstrate that the SDM model could be reduced to SLM and SEM models. Both null hypotheses were rejected and passed the significance test. As a result, this study combines the aforementioned analyses to select the SDM model under double-fixed effects as the final interpretation model for spatial regression.

**Table 3.** Statistical test results for model selection.

| Type of Test | Null Hypothesis | Statistic | *p*-Value |
|---|---|---|---|
| Hausman test | The individual effect has no correlation with the regression variable | 36.59 | 0.000 |
| LR test | Spatial fixed-effect nested within double fixed-effect | 68.82 | 0.000 |
| | Time fixed-effect nested within double fixed-effect | 651.55 | 0.000 |
| LM-Spatial error | No spatial correlation between error terms | 21.65 | 0.000 |
| Robust LM-Spatial error | | 22.81 | 0.000 |
| LM-Spatial lag | No spatial correlation between lag terms | 13.61 | 0.000 |
| Robust LM-Spatial lag | | 14.77 | 0.000 |

**Table 4.** Wald test and LR rest (double fixed-effect).

| Statistical Tests | SLM vs. SDM | | SEM vs. SDM | |
|---|---|---|---|---|
| | Z-Value | *p*-Value | Z-Value | *p*-Value |
| Wald test | 51.82 | 0.000 | 51.60 | 0.000 |
| LR test | 48.01 | 0.000 | 47.39 | 0.000 |

*4.4. Spatial Regression Results*

This study used the economic distance weight matrix ($W_1$) and the SDM model to investigate the geographical effects of a GBE on sustainable economic development. For the robustness test, the geographic distance weight matrix ($W_2$) was also utilized. Table 5 presents the outcomes. With the exception of human capital, all of the control variables were statistically significant. According to the coefficient of GBE, which was 0.1026 and significant at 1%, there is a commensurate improvement in sustainable economic growth of 0.1026% for every 1% increase in the green business environment.

On one hand, creating a supportive environment for green businesses can lead to reduced operational costs, encourage innovation, and expand the market. On the other hand, it may also attract more foreign investment, which can boost long-term economic growth. Furthermore, optimizing a green business environment as an "economic neighbor" has a positive impact on the sustainable economic development of the province or municipality, as evidenced by the coefficient of spatial interaction (***W*** *\*lnGBE*) of 0.2362, which is statistically significant at the 1% level.

All geographical interactions between the control variables were also significant, with the exception of foreign direct investment. Notably, the spatial weight coefficient was −0.0296 and significant at 1%, while the coefficient of the primary influence of environmental regulation was −0.0069 and exhibited 5% statistical significance. It is conceivable that environmental rules, which could operate as external impediments to economic expansion, are to blame because they have a direct bearing on expenditure and cost. More specifically, excessive environmental protection costs will harm sustainable economic growth.

The spatial regression results from the SDM model are displayed in Table 5 together with the geographic distance weight matrix ($W_2$) for robustness testing. The results obtained using the economic distance weight matrix were consistent with the main effect and geographical weight coefficients of the GBE, which were 0.0717 and 0.5582, respectively, and significant at 5% and 1% for the core explanatory factors. Using geographic distance as the spatial weight matrix also resulted in a greater significance of the positive geographical spillover effect than the previously suggested model for spatial effect. This can show how the spatial spillover effect of the GBE had a major influence on the provinces and municipalities that are nearest to one another geographically. The robustness of the empirical findings was demonstrated by the control variable coefficients, which had nearly the same signs and significance as the preceding ones.

**Table 5.** Results of spatial econometric model.

| Variable | OLS | Spatial Weight Matrix (W$_1$) | Spatial Weight Matrix (W$_2$) |
|---|---|---|---|
| | Model I | Model II | Model III |
| lnGBE | 0.6573 *** | 0.1026 *** | 0.0717 ** |
| | (0.070) | (0.031) | (0.030) |
| lnFC | 0.2766 *** | 0.0655 *** | 0.0892 *** |
| | (0.069) | (0.014) | (0.014) |
| lnHC | 2.4981 *** | −0.0039 | 0.0946 |
| | (0.223) | (0.102) | (0.100) |
| lnFDI | −0.0145 | 0.0106 *** | 0.0068 * |
| | (0.017) | (0.004) | (0.004) |
| lnER | 0.0022 | −0.0069 ** | 0.0013 |
| | (0.018) | (0.003) | (0.003) |
| lnIS | 0.4112 *** | −0.1658 *** | −0.1390 *** |
| | (0.112) | (0.039) | (0.037) |
| W*lnGBE | | 0.2362 *** | 0.5582 *** |
| | | (0.084) | (0.207) |
| W*lnFC | | −0.0666 * | 0.5013 *** |
| | | (0.039) | (0.107) |
| W*lnHC | | 0.6280 ** | 1.8581 ** |
| | | (0.278) | (0.769) |
| W*lnFDI | | 0.0151 | −0.0935 ** |
| | | (0.011) | (0.037) |
| W*lnER | | −0.0296 *** | 0.0635 *** |
| | | (0.008) | (0.023) |
| W*lnIS | | 0.4800 *** | −0.4939 |
| | | (0.113) | (0.341) |
| ρ | | 0.2095 ** | 0.4866 *** |
| | | (0.103) | (0.140) |
| N | 300 | 300 | 300 |
| R$^2$ | 0.7166 | 0.7892 | 0.3194 |
| Log-L | | 674.8425 | 686.8714 |

Note: *, **, and *** indicate significance at the 10%, 5%, and 1% levels, respectively—standard errors are in parentheses.

### 4.5. Spatial Effect Decomposition

The spatial model is capable of identifying the unique characteristics of spatial units in different provinces (municipalities) [20]. However, when spatial autocorrelation is considered, the explanatory variable coefficients in the SDM model may not accurately reflect the impact of independent factors on dependent variables [21]. In order to evaluate the spatial spillover effects, it is important to determine the direct, indirect, and total effects. Specifically, the indirect effect refers to the possible spillover effect that independent factors may have on sustainable economic development, while the direct effect represents the impact of changes in independent variables on sustainable economic growth within a fixed spatial unit.

Table 6 demonstrates that a green business environment has a direct and significantly positive impact on local sustainable economic development under the economic distance weight matrix (Model II). This implies that optimizing a green business environment can effectively raise the standards of local economic development. A convenient business environment can help to attract and nurture elite individuals and businesses. By promoting clustering and technology spillover effects, enhancing the green business environment can lead to high-quality development and economic sustainability. This eventually results in improved product quality and competitiveness. Moreover, strengthening the green business environment has a significant and positive indirect effect. This shows that the positive spatial spillover effects enhance the economic sustainability of municipalities and their neighboring provinces.

Both the direct and indirect effects of the green business environment were notably positive in Model III with the geographic distance weight matrix. This demonstrates how enhancing the environment for green business benefits nearby neighborhoods as well as the sustainable economic development of the region. It is important to note that Model III had an indirect effect on the green business environment, which was higher than Model II's coefficient. The coefficient for Model III was 1.991, while Model II's coefficient was 0.3162. This means that the spatial effects of the green business environment have a greater impact on provinces and municipalities that are closer in terms of geography than in terms of the economy. In other words, the closer provinces and municipalities are to each other, the more significant their influence on sustainable economic development becomes due to their increasing influence of the green business environment. As the geographic distance between provinces and municipalities increases, the spatial spillover effect gradually decreases.

**Table 6.** Result of spatial effect decomposition.

| Variable | Model II | | | Model III | | |
|---|---|---|---|---|---|---|
| | Direct Effect | Indirect Effect | Total Effect | Direct Effect | Indirect Effect | Total Effect |
| lnGBE | 0.1122 *** | 0.3162 *** | 0.4284 *** | 0.0972 *** | 1.1991 ** | 1.2963 ** |
| | (0.031) | (0.108) | (0.117) | (0.032) | (0.532) | (0.544) |
| lnFC | 0.0618 *** | −0.0754 * | −0.0136 | 0.1105 *** | 1.0723 *** | 1.1828 *** |
| | (0.012) | (0.043) | (0.046) | (0.017) | (0.365) | (0.377) |
| lnHC | 0.0248 | 0.7588 ** | 0.7836 ** | 0.1794 | 3.7181 ** | 3.8975 ** |
| | (0.111) | (0.319) | (0.350) | (0.123) | (1.834) | (1.898) |
| lnFDI | 0.0116 ** | 0.0221 | 0.0337 * | 0.0035 | −0.1754 ** | −0.1719 * |
| | (0.005) | (0.016) | (0.018) | (0.005) | (0.087) | (0.091) |
| lnER | −0.0086 ** | −0.0376 *** | −0.0462 *** | 0.0036 | 0.1326 * | 0.1362 * |
| | (0.004) | (0.011) | (0.012) | (0.004) | (0.073) | (0.075) |
| lnIS | −0.1470 *** | 0.5514 *** | 0.4044 ** | −0.1598 *** | −1.1191 | −1.2789 * |
| | (0.040) | (0.155) | (0.172) | (0.044) | (0.706) | (0.734) |

Notes: *, **, and *** indicate significance at the 10%, 5%, and 1% levels, respectively.

*4.6. Regional Heterogeneity Analysis*

This study aims to examine how the spatial spillover effect of the GBE affects the sustainable development of the regional economy in China. We consider the varying economic foundations and resource allocation across China's eastern, central, and western regions. Table 7 provides a detailed breakdown of the spatial effect and the estimation results.

The research shows that a green business environment (lnGBE) has a positive and direct impact on sustainable economic development (0.1268 **), particularly in the eastern regions. This means that improving the local GBE will positively impact sustainable economic development. However, the indirect effect of GBE (0.1442) is insignificant, which suggests that improvements to the green business environment in neighboring municipalities or provinces will not have an impact on sustainable economic development. To put it simply, a focus on improving the GBE in the Eastern region will primarily benefit the long-term growth of the local economy.

In contrast, both direct and indirect benefits in the central and western regions were noteworthy, at 1%. The improvement of the GBE in these two regions, as well as in their neighboring regions, has a significant impact on sustainable economic development, especially in comparison to other regions. These findings align with some research [22,23] regarding the effects of various regional business environments on sustainable economic development. Overall, it can be concluded that the development of a green business environment has a positive impact on sustainable economic growth, and this impact is more significant in regions with less-developed business environments.

**Table 7.** Heterogeneity analysis of spatial effects in various regions.

| Region | Variable | Direct Effect | Indirect Effect | Total Effect |
|---|---|---|---|---|
| Eastern Region | lnGBE | 0.1268 ** (0.051) | 0.1442 (0.108) | 0.2710 *** (0.100) |
| | lnFC | 0.0761 *** (0.028) | 0.2134 *** (0.062) | 0.2895 *** (0.065) |
| | lnHC | 0.0572 (0.183) | 0.1390 (0.331) | 0.1963 (0.283) |
| | lnFDI | 0.0041 (0.010) | −0.0582 ** (0.027) | −0.0541 * (0.030) |
| | lnER | −0.0125 ** (0.006) | −0.0068 (0.011) | −0.0193 * (0.011) |
| | lnIS | −0.0650 (0.097) | −0.0378 (0.257) | −0.1028 (0.223) |
| Central Region | lnGBE | 0.1629 *** (0.032) | 0.1902 ** (0.090) | 0.3532 *** (0.093) |
| | lnFC | 0.1525 *** (0.014) | −0.0641 (0.068) | 0.0884 (0.070) |
| | lnHC | −0.0101 (0.121) | 0.2935 (0.240) | 0.2834 (0.277) |
| | lnFDI | 0.0517 *** (0.009) | 0.1146 ** (0.047) | 0.1663 *** (0.054) |
| | lnER | 0.0071 (0.005) | 0.0243 ** (0.010) | 0.0314 *** (0.011) |
| | lnIS | −0.4155 *** (0.048) | −0.1687 (0.203) | −0.5842 ** (0.235) |
| Western Region | lnGBE | 0.1622 *** (0.055) | 0.4739 *** (0.124) | 0.6360 *** (0.148) |
| | lnFC | 0.0365 ** (0.016) | −0.1769 *** (0.045) | −0.1404 *** (0.049) |
| | lnHC | −0.1521 (0.157) | 0.2284 (0.360) | 0.0763 (0.423) |
| | lnFDI | −0.0002 (0.005) | 0.0314 *** (0.011) | 0.0312 ** (0.013) |
| | lnER | 0.0051 (0.006) | 0.0126 (0.012) | 0.0177 (0.014) |
| | lnIS | −0.1319 * (0.069) | −0.1429 (0.184) | −0.2747 (0.226) |

Notes: *, **, and *** indicate significance at the 10%, 5%, and 1% levels, respectively.

The provinces in the eastern region have a significant and positive impact on fixed capital (lnFC), both directly and indirectly. This indicates that fixed capital in these areas can significantly contribute to sustainable economic development. On the other hand, the central provinces can invest in other provinces while also promoting their own sustainable economic development. However, the western provinces mainly receive fixed capital from other provinces, which negatively impacts sustainable economic development. One possible explanation is that higher fixed investments use more natural resources.

Foreign direct investment (lnFDI) has different effects on different regions of the country. In the eastern provinces, FDI has a negligible direct effect, while the indirect effect is significantly negative. This implies that the eastern provinces primarily invest in other provinces, which does not contribute to their sustainable economic development, but instead drains their resources. In contrast, the central provinces experience both significant and positive direct and indirect effects, indicating that they can significantly improve their sustainable economic development regardless of the source of the FDI. Finally, the western provinces welcome FDI from other regions, as it helps them to strengthen their economy and promote sustainable development.

Environmental regulations (InER) can have a direct impact on sustainable development, and it may not always be positive. When environmental regulations increase the costs of running a business, it can hinder the sustainable economic development of the eastern provinces. On the other hand, the central provinces' environmental regulations have a positive indirect effect, indicating that the current regulations in those regions can be maintained. Excessive environmental regulations can do more harm than good, and it is important to strike a balance between business interests and environmental concerns.

Results regarding industrial structure (lnIS) indicate that, even though the direct effects are considerably negative in the western and central provinces, the indirect effects do not hold much significance in the eastern provinces. This implies that despite making efforts to improve their industrial structure in comparison to other provinces, the western and central provinces are not contributing to the sustainable economic development of their own provinces. Additionally, neither the eastern nor the central provinces possess significant levels of human capital (lnHC).

## 5. Conclusions

### 5.1. Provincial and Municipal Business Environment Evaluation

This study applies principal component analysis (PCA) to assess the green business environments across different provinces and municipalities in mainland China, spanning from 2011 to 2020. The evaluation indices for the green business environment were proposed and referenced from other business environment evaluation indices published by various organizations. Furthermore, the regions of eastern, central, and western China are taken into account to categorize the provinces and municipalities, and their respective green business environments are analyzed.

According to the findings, the green business environment rankings place Guangdong, Shanghai, Jiangsu, Beijing, and Zhejiang in the top five, and Xinjiang, Guizhou, Ningxia, Gansu, and Qinghai at the bottom. Additionally, the eastern regions' provinces and municipalities exhibit superior performance in terms of the green business environment when compared to those in the central and western regions.

### 5.2. Effect of GBEEI on Sustainable Economic Development

The development of a green business environment holds great potential for fostering sustainable and high-quality economic growth. Moreover, such growth can extend beyond individual provinces and municipalities, benefiting neighboring regions as well. To investigate this phenomenon, the Spatial Dubin Model was employed to examine the spatial spillover effect of the green business environment on sustainable economic development across 30 provinces and municipalities between 2011 and 2020. Both economic and geographic distance weight matrices were utilized in the analysis.

According to the research, the green business environment has a significant impact on sustainable economic development in its region and neighboring areas. This dissertation also delves into the variations in the eastern, central, and western regions of China. While the eastern region solely relies on its own influence, the green business environments of both the region and its neighbors contribute to sustainable economic growth in the central and western regions.

### 5.3. Research Limitations and Future Research

In summary, the suggested index for evaluating the green business environment in mainland China is well-suited for the task at hand. Additionally, given its status as a central explanatory variable, the green business environment plays a significant role in promoting sustainable economic growth and corporate social responsibility, thus offering valuable insights for policymakers and businesses alike. Moving forward, there are several areas where this research can be further developed and expanded upon.

First, new indicators can still be considered for future research, even if the Green Business Environment Evaluation Index has been enhanced and supplemented in light of

earlier iterations. For instance, water pollution is one of the primary causes. Numerous studies have evaluated the emission of water pollution and its monitoring, control, eco-remediation, and potential environmental impact; among these is the remarkable study by Chen et al. that shows the impact of water pollution on ecosystem health and human health [24]. Our research's shortcoming, meanwhile, is the dearth of information on the emissions of water pollution. Since water contamination causes two million deaths a year, it has grown to be a severe worldwide problem that needs careful consideration.

Furthermore, the analysis of the study's spatial spillover impact involves incorporating both economic and geographic distance weight matrices. While these two weight matrices are commonly used, alternative weighting matrices can also be explored to validate the findings. Additionally, future research can adopt diverse methodologies to assess the economic distance between provinces and municipalities, such as examining their similarities, and can also incorporate geographical adjacency as a means of measuring distance.

In further research, we attempt to utilize the Fuzzy-set Qualitative Comparative Analysis (fsQCA) method, which has gained recognition for its objective and statistically informed approach to deriving predictive conclusions. This exemplifies a more asymmetrical way of thinking, in line with complexity theory [25]. The fsQCA methodology takes a comprehensive approach, considering both qualitative analysis and grouping perspectives. It regards the object of study as a collection of different combinations of condition variables and utilizes ensemble analysis to determine the collective relationships between groups and outcomes. This is particularly useful in tackling complex causal questions involving concurrent causality, causal symmetry, and scenario equivalence.

**Author Contributions:** Conceptualization, C.-W.L.; methodology, H.-H.H.; software, C.-C.W.; formal analysis, P.K.; investigation, P.K.; resources, C.-C.W.; data curation, P.K.; writing—original draft preparation, P.K.; writing—review and editing, C.-W.L.; supervision, C.-W.L.; project administration, H.-H.H. funding acquisition, C.-C.W. All authors have read and agreed to the published version of the manuscript.

**Funding:** This research received no external funding.

**Institutional Review Board Statement:** Not applicable.

**Informed Consent Statement:** Informed consent was obtained from all subjects involved in the study.

**Data Availability Statement:** Data are contained within the article.

**Acknowledgments:** We would like to thank all the participants in this study.

**Conflicts of Interest:** The authors declare no conflict of interest.

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
