# Peer review of "The Assessment of Green Business Environments Using the Environmental–Economic Index: The Case of China"

_sustainability, doi:10.3390/su152316419_

Round 1
Reviewer 1 Report
Comments and Suggestions for Authors
This article is a study on the use of environmental economic indicators to evaluate the green business environment, with a focus on China. This study evaluated the business environment of 30 provinces and cities in China, combined with ecological and environmental protection and sustainable development indicators, and constructed a new green business environment index. At the same time, the impact of the index on macroeconomic sustainable development and micro enterprise operations was also analyzed, providing valuable insights into the country's capabilities and competitiveness. This article needs to address several minor issues before publication.
1. In Table 1, why only analyze NOx emissions in the X54 Air Pollution in the Green Environment
Without analyzing the emissions of other greenhouse gases? Gases such as carbon dioxide and methane chloride can also cause atmospheric pollution.
2. The captions of the vertical and horizontal axes in Fig. 2 are not the same as the distance between the coordinate axes (similar to other images), and the viewing effect for readers is not good!
3. Why is the P-value of Table 2 and Table 3 0? This seems to be statistically unreasonable.
4. The R2 value in Table 4 is relatively small (with a maximum of only 0.7892), does this mean that the fit of the data is not high, and thus the credibility of the conclusions obtained may not be high?
Comments on the Quality of English Languagegood
Author Response
We have revised my manuscript according to your comments. Please refer to the attachment.

Reviewer 2 Report
Comments and Suggestions for Authors
Lei estimated the green business environment index in 30 provinces of China using eigenvalues obtained from principal components analysis. Then, the study compared the overall index of green business environment across the eastern region, central region, and western region. There are several flaws in the manuscript. I can not recommend the publication in the current form.
Table 1. The study should include more items on green environment. The current study only includes NOx emission as the metric of air pollution. The study should include PM2.5, SO2, and CO data, which are available at the Ministry of Ecology and Environment of the People’s Republic of China. Furthermore, the dataset of water pollution also should be included for the metrics of a Green Environment. Otherwise, the estimates of the green business environment index may lead to the bias of the green business environment index in individual provinces.
The methods for estimating the green business environment index are similar to the human development index proposed by UNDP(http://hdr.undp.org/en/humandev). Song and Mei estimated the human development index in 30 provinces of China from 2011 to 2018. The trends in the ranking of the human development index from Song and Mei are not consistent with the results in this study. The authors should illustrate the differences in findings between the two studies, which may be ascribed to the absence of some metrics of Green Environment such as the dataset of water pollution.
The major flaw is the method section. The method section should include the details of the estimates of the green environment index. The current details are not clear.
The method section also should include the details of several statistical tests including the Hausman test, LR test, LM-Spatial error, etc.
The study should include the map of 30 studied provinces in China and regions affiliated with the Eastern region, Central region, and Western region, respectively. In doing so, the readers could digest the results from individual provinces more easily.
The manuscript should re-organize the introduction of the manuscript and highlight the purpose of the study. Why is the study carried out to estimate the green business environment index? What is the new information from the green business environment index compared with another composite index such as the human development index?
The references are not adequate.
Song, Y., Mei, D. Sustainable development of China's regions from the perspective of ecological welfare performance: an analysis based on GM(1,1) and the Malmquist index.
Author Response
Thank you very much for your suggestion. We have already revised my manuscript according to your comments. Please refer to the attachment.

Reviewer 3 Report
Comments and Suggestions for Authors
This study examines the impact of the green business environment on sustainable economic development in different areas. For this purpose a novel green business environment index is constructed and analyzed for its impact on macroeconomic sustainable development and micro-enterprise operation. Thus, it is congruent with the aims and scope of Sustainability.
The research gap is clearly presented.
The abstract is clear, concise and interesting. From the beginning of the manuscript, author(s) show a clear research question.
The novelty of research undertaken is that it introduces the GBE assessment index, which is a combination of previous evaluation indexes and a new ecological environment evaluation index that is tailored to Chinese provinces. To take into account spatial charecteristics of the territory the author(s) use global space autocorrelation and local space autocorrelation.
The manuscript is easy to read, and its structure is consistent with the scientific style.
Conclusions made are thoroughly supported by the results presented in the article. There are also convincing concluding debate regarding the unique contribution of GBEEI on Sustainable Economic Development.
Research limitations and future research directions are also presented in this section.
From my point of view, this manuscript can be accepted in the current form.
Thank you for the opportunity to review this article and good luck!
Author Response
Thank you very much for your encouragement.

Round 2
Reviewer 2 Report
Comments and Suggestions for Authors
Accept
Author Response
I would like to express my sincere gratitude for your valuable feedback on my paper. Your comments have provided me with valuable insights and suggestions to make my work more impactful and useful to readers. Thank you for taking the time to share your thoughts with me.